# Is There Any Difference in the In Situ Immune Response in Active Localized Cutaneous Leishmaniasis That Respond Well or Poorly to Meglumine Antimoniate Treatment or Spontaneously Heal?

**DOI:** 10.3390/microorganisms11071631

**Published:** 2023-06-22

**Authors:** Jéssica Leite-Silva, Carla Oliveira-Ribeiro, Fernanda Nazaré Morgado, Maria Inês Fernandes Pimentel, Marcelo Rosandiski Lyra, Aline Fagundes, Luciana Freitas Campos Miranda, Claudia Maria Valete-Rosalino, Armando Oliveira Schubach, Fátima Conceição-Silva

**Affiliations:** 1Laboratory of Immunoparasitology, Oswaldo Cruz Institute (IOC), Fundação Oswaldo Cruz (Fiocruz), Rio de Janeiro 21041-250, RJ, Brazil; jessicaleite-@hotmail.com (J.L.-S.); morgado@ioc.fiocruz.br (F.N.M.); 2Service of Oncological Dermatology—National Institute of Cancer (INCA), Rio de Janeiro 20570-120, RJ, Brazil; carla_ribb@yahoo.com.br; 3Laboratory of Clinical Research and Surveillance in Leishmaniasis (LAPCLIN VIGILEISH) National Institute of Infectology Evandro Chagas (INI), Fiocruz Rio de Janeiro 21041-250, RJ, Brazil; minesfpimentel@gmail.com (M.I.F.P.); marcelolyradermato@hotmail.com (M.R.L.); aline.fagundes@ini.fiocruz.br (A.F.); aschubach@yahoo.com (A.O.S.)

**Keywords:** host-parasite interaction, immunopathology, cutaneous leishmaniasis, spontaneous healing, treatment relapse, in situ immune response, macrophages, cell-mediated immunity, immunohistochemistry, CD163

## Abstract

Localized cutaneous leishmaniasis caused by *Leishmania braziliensis* can either respond well or poorly to the treatment or heal spontaneously; It seems to be dependent on the parasite and/or host factors, but the mechanisms are not fully understood. We evaluated the in situ immune response in eighty-two active lesions from fifty-eight patients prior to treatment classified as early spontaneous regression (SRL-n = 14); treatment responders (GRL-n = 20); and non-responders (before first treatment/relapse, PRL1/PRL2-n = 24 each). Immunohistochemistry was used to identify cell/functional markers which were correlated with the clinical characteristics. PRL showed significant differences in lesion number/size, clinical evolution, and positive parasitological examinations when compared with the other groups. SRL presented a more efficient immune response than GRL and PRL, with higher IFN-γ/NOS2 and a lower percentage of macrophages, neutrophils, NK, B cells, and Ki-67+ cells. Compared to SRL, PRL had fewer CD4+ Tcells and more CD163+ macrophages. PRL1 had more CD68+ macrophages and Ki-67+ cells but less IFN-γ than GRL. PRL present a less efficient immune profile, which could explain the poor treatment response, while SRL had a more balanced immune response profile for lesion healing. Altogether, these evaluations suggest a differentiated profile of the organization of the inflammatory process for lesions of different tegumentary leishmaniasis evolution.

## 1. Introduction

Leishmaniasis is an infectious disease mainly distributed in tropical and sub-tropical regions with varied clinical presentation and the potential to cause deformities and death. The main clinical presentations are visceral (VL) and tegumentary leishmaniasis (TL); the latter of which can be subdivided into cutaneous (CL) and mucosal (ML) leishmaniasis. In the last two decades, more than 1,000,000 cases of TL were reported to the Pan American Health Organization (PAHO), with a yearly average greater than 50,000 cases. In Brazil, TL is mainly caused by *Leishmania* (*Viannia*) *braziliensis* and the most common clinical presentation is termed localized cutaneous leishmaniasis (LCL) [1,2,3,4,5]. LCL is characterized by one or a few painless ulcers in a single segment of the body, with infiltrated erythematous-violaceous raised borders and a base containing granulations with a small amount of purulent discharge. Cases of early spontaneous healing without treatment can occur and have been identified during diagnosis; these cases are termed early spontaneous regression [6]. Some hypotheses for this phenomenon are related to the early establishment of a specific immune response that is efficient enough to eliminate the parasite [7,8,9].

Of the cases that demand specific treatment, some patients do not reach lesion healing after one complete treatment cycle, which is characterized as therapeutic relapse. The phenomenon of non-response to primary treatment has been increasingly reported in the literature and is a great concern because the therapeutic arsenal for leishmaniasis is still quite restricted [10,11,12,13,14,15]. Research groups have suggested that such occurrences of therapeutic relapse could be due to different factors including patient weight, number of lesions, co-infections, treatment regimens, parasite virulence, and whether the parasite strain has developed resistance to the specific treatment [11,12,14,16,17,18]. Gagini et al. [19] evaluated parasite isolates before treatment with meglumine antimoniate and when relapse was characterized, and the results showed that even genetically similar isolates responded differently to meglumine antimoniate. This suggests that factors other than parasite genetics could have an influence on the therapeutic outcome in patients that have a poor response to treatment. Furthermore, Baptista et al. [20] demonstrated that low-dose or intralesional treatment did not induce in vitro resistance to antimonials. As yet, there is no conclusive explanation for therapeutic relapse, making it difficult to identify cases with the potential for poor therapeutic response.

The literature shows that spontaneous healing or the development of severe lesions in TL is influenced by the immune response and characteristics of the organized inflammation in the lesion sites [21,22,23,24]. In the murine model, ulcer formation has been suggested as being a consequence of the inflammatory process and not necessarily caused by the parasite load [25]. In humans, in situ studies of typical LCL lesions revealed the predominance of T lymphocytes, macrophages, and neutrophils [26,27,28,29], and the proportion of these cells seems to influence the lesion evolution. Some cytokines are also essential to the evolution of TL lesions to healing; In addition to the well-known influence of immune responses with a predominance of Th1 T lymphocytes (favoring infection control) or Th2 (favoring potential parasite survival and consequently maintenance of active infection), other populations of lymphocytes such as T helper 17 (Th17) may also be present. Interferon (IFN)-γ, for instance, stimulates macrophages to increase the activity of the enzyme nitric oxide synthase 2 (NOS2), consequently, increasing the production of nitric oxide (NO), as well as superoxide radicals, which are the main mediators of parasite death [30,31]. Infection with *Leishmania* species of the *Viannia* subgenus also triggers macrophage activation. Macrophages are highly plastic cells, showing a spectrum of phenotypes depending on the stimulus received from the environment. Macrophages have enormous plasticity and can perform diverse functions depending upon their phenotypic state, but for didactic purposes, they have been classified as Classically activated (M1) macrophages or alternatively activated (M2) macrophages [31]. M1 favors parasite control while M2 cells favor parasite growth and survival, consequently, maintaining the lesion [32,33,34,35]; thus, the proportions of these macrophage activation types could have an effect on lesion evolution. The presence of other cell types such as T helper 17 (Th17) and T regulatory (T reg) cells could also influence the disease evolution towards healing [36]. Extracellular trap (ET) formation, as a mechanism to control parasite load, mainly by neutrophils (NETs), has also been demonstrated in TL [28,37,38,39].

A few studies have assessed and compared the immune response in cases of CL with early spontaneous healing and those that respond poorly to treatment [6,40,41,42,43,44]. The use of in situ skin inflammatory reaction tools is important to identify the structure and organization of the in situ inflammatory process in TL and other diseases. Such information allows the identification of the infectious agent and the type of cells, cytokines, and functional markers involved during the inflammatory response [23,45,46,47,48]. Studies with these tools have demonstrated that interleukin (IL)-17 is involved in lesion chronicity [42]. Maretti-Mitra et al. [41] observed greater IFN-γ, IL-10, and tissue growth factor (TGF)-β expression in the lesions of patients who evolved with non-response to treatment. Despite the recognition of the role played by cytokines, the mechanism by which infection control or therapeutic relapse occurs is still not fully understood [23,49,50,51,52]. Therefore, characterizing and comparing the type and function of the cells involved in lesions that heal or relapse is important to understand different clinical characteristics of TL.

The purpose of this study was to evaluate and compare some aspects of the in situ immune response in the active cutaneous lesions of patients presenting early spontaneous regression of TL (without treatment) and responders or non-responders to specific treatment with meglumine antimoniate (prior to treatment and at therapeutic relapse). These findings were then correlated with clinical and parasitological parameters in order to identify possible related patterns.

## 2. Materials and Methods

### 2.1. Patients

Eighty-two active lesions from fifty-eight patients with the localized cutaneous form of TL, as confirmed by parasite detection (culture, PCR, histopathology and/or immunohistochemistry), were selected and classified into three groups: (A) lesions that evolved early spontaneous regression during the diagnosis procedure and consequently no treatment was required (SRL; n = 14); (B) “responders to meglumine antimoniate treatment” with no relapse for at least 1 year of longitudinal monitoring (GRL; n = 20); and (C) “non-responders to meglumine antimoniate treatment” (PRL; further divided as initial lesion—PRL1; n = 24, and relapse—PRL2; n = 24). The specific treatment comprised the use of meglumine antimoniate administered intramuscularly at 5 mg Sb5+/kg/day for 30 days as previously described [13,53,54]. Patients were monitored on the 15th day of treatment, at the end of treatment (30 days), monthly up to 3 months post-treatment, and then at 6- and 12-months post-treatment. Cases of worsening or lesion reactivation within the first three months after treatment were considered “non-responders to primary meglumine antimoniate treatment” (PRL). In this group, each patient was examined and the lesions were biopsied on two occasions: at the time of the initial lesion (PRL1) and at the time when therapeutic relapse was identified (PRL2). All patients underwent otolaryngology examination during the follow-up period and no signs of mucosal lesions were detected in any patient. Patients under 15 and over 80 years old were excluded due to the possibility of physiological changes in the immune response. Comorbidities that could significantly alter the quality of the immune response (HIV, chronic use of corticosteroids, decompensated diabetes mellitus, neoplasms, autoimmune diseases, hepatitis), as well as patients, that for whatever reason did not sign the informed consent form, were also excluded. The patients had no previous history of infection with parasites of the genus *Leishmania*. The study was approved by the institutional Research Ethics Committees of the Instituto Oswaldo Cruz and Instituto Nacional de Infectologia Evandro Chagas (CEP-IOC: CAEE 88890518.6.0000.5248 and CEP-INI: CAEE 88890518.6.3001.5262). The variables of interest for the characterization of the study group were sex, age, size and number of lesions, lesion location and its characteristics, and response to the treatment. When possible, the *Leishmania* species, isolated by the culture of the tissue obtained during the biopsy, was characterized using multilocus enzyme electrophoresis (MLEE), PCR of HSP-70, restriction fragment length polymorphism (RFLP) with HaeIII and BstUI enzymes, and/or genetic sequencing [55].

### 2.2. Immunohistochemistry

The immunohistochemistry technique was carried out on 3 to 4-μm sections of cryopreserved tissue from the lesions affixed to silanized slides as previously described [26,43]. After a set of blockage steps, primary antibodies were added and the sections were incubated for 12–16 h at 4 °C in order to perform a phenotypic (cell type) and functional (enzymes, surface molecules, and cytokines) characterization of the inflammatory infiltrate, using antibodies against CD4 (clone 4B12), CD8 (clone 144B), CD22 (clone 4Kb128), neutrophil elastase (clone NP57), mast cell (clone AA1), CD56 (clone 123C3), CD25 (clone BC96), Ki-67 (clone MIB-1), CD68 (clone EBM11) (all obtained from Dako-Carpinteria, CA, USA), CD163 (clone EDHU-1; BioRad- Hercules, CA, USA), CD206 (clone 15-2; BioRad), NOS2 (iNOS; clone ab3523; Abcam- Waltham, Boston- USA), IFN-γ (clone B27; BD Biosciences Pharmingen-San Diego, CA, USA), FoxP3 (clone PCH101; BD Biosciences Pharmingen), ST2L (clone B4E6; MD bioscience- Oakdale, MN, USA), IL-33 (clone 390412; R&D Systems-Minneapolis, MN-USA), and anti-*Leishmania* sp. (a kind gift from Dr. MF Madeira, INI-Fiocruz-Rio de Janeiro, Brazil). The sections were incubated with biotinylated secondary antibody (Zymed Laboratories Inc.-San Francisco, CA, USA), stained using the Histostain^®^-Plus enzyme labeling kit containing streptavidin peroxidase (Invitrogen-Carlsbad, CA, USA) and then the AEC staining kit (Zymed Laboratories Inc.), with intermediate washing steps in PBS. Following this sequence, the sections were stained with Mayer’s hematoxylin (Dako), then a coverslip was added with Faramount Mounting Medium (Dako). Negative controls without primary antibodies were carried out. Slides were analyzed on an optical microscope and the percentage of labeled cells was determined by counting 500 cells or 10 fields (1000× magnification). For NOS2 specifically, label intensity was determined by the number of positive sites per 20× field: discrete (1 positive site), moderate (2 positive sites), intense (3 positive sites), and very intense (4 a 5 positive sites), as previously described [26]. The pictures were stored on the Motic Images Plus program (version 2.0, Motic China Group Co. Ltd., Hong Kong, China).

### 2.3. Immunofluorescence

To enable double labeling, immunofluorescence was performed following the same steps as conventional immunohistochemistry regarding the application of the primary antibodies. Subsequently, sections were stained with specific fluorophore-conjugated secondary antibodies, Alexa Fluor 488 (Invitrogen), R-phycoerythrin (PE; Invitrogen), and DyLightTM 633 (ImmunoReagents-Raleigh, NC, USA). Fluoromount-G Mounting Medium with DAPI (Invitrogen) was used to mount the slide and coverslip. Images were captured on a Zeiss Axio lmager M1 microscope with the AxioVision software (Carl Zeiss Microscopy, LLC, White Plains, NY, USA).

### 2.4. Statistical Analysis

A specific database was constructed for this study using the R 4.2.1 software (R Project for Statistical Computing, The R Foundation, Vienna, Austria) and GraphPad Prism 8.0 (Dotmatics, Woburn, MA, U.S.A.). The Shapiro–Wilk test was used to verify the normality of the data. Analysis of immunohistochemistry results of the active primary lesions was compared two-by-two using the Mann–Whitney non-parametric test, while the Wilcoxon test was used for samples paired in the same patient (PRL). The Kruskal–Wallis test was used for the analysis and characterization of patient data and comparison of three or more groups using Dunn’s Test correction. The chi-square exact test was used to analyze NOS2 expression. Data were presented in distribution as the mean ± SEM (minimum error) or as the median. When applicable, *p* values ≤ 0.05 were considered significant.

## 3. Results

### 3.1. PRL Patients Had More Severe Skin Lesions and Showed Greater Positivity in the Parasite Detection Tests

There were significant differences in the clinical parameters between the groups. PRL cases presented the highest number of lesions (*p* = 0.032) and greatest lesion size (*p* = 0.05), which varied from 5 to 90 mm in diameter (Table 1). Considering all patients, lower limbs were the most affected (37.68%), followed by upper limbs (34.78%), trunk (15.94%), neck (5.17%), and head (5.17%). The diagnosis of leishmaniasis was confirmed through parasite detection in at least one of the three/four methods performed (culture, PCR, immunohistochemistry and/or histopathology). Patients presenting early spontaneous healing lesions (SRL) mostly showed parasitological confirmation with only one positive test, that being PCR in the majority of cases (85.71%); conversely, in the poor responder group (PRL), 91.67% of patients had three or more positive exams (*p* < 0.0001, Table 1). In 39 patients (67.24% of the total), it was possible to characterize the parasite at the species level; in all of these cases, the causative species was identified as *L. braziliensis*. Of these 39 patients, 23 were PRL, 13 were GRL, and 3 were SRL.

### 3.2. Different In Situ Cellular Profiles Are Correlated with Patient Characteristics

Staining of the biopsy sections of the initial active lesions of the groups (SRL, GRL, and PRL1), as well as the lesion at therapeutic failure (PRL2) showed significant differences in all cell and functional markers, except for CD8, mast cell tryptase, and CD206. Table 2 shows the percent distribution of the markers evaluated in each group and Figure 1, Figure 2, Figure 3, Figure 4 and Figure 5 show the comparative statistical analyses among the groups. Appendix A demonstrate the comparison of the two moments of PLR patients using the Wilcoxon test.

### 3.3. Distribution of CD4+ and CD8+ T Lymphocytes and CD22+ B Lymphocytes

CD4+ and CD8+ T lymphocytes presented homogenous distribution throughout the dermis in all groups. PRL had fewer CD4+ T cells than SRL specifically, while GRL was found to have significantly more CD4+ cells compared with the lesions of the other groups (*p* < 0.0001; Figure 1), no statistically significant differences were observed in the groups regarding CD8+.

**Figure 1 microorganisms-11-01631-f001:**
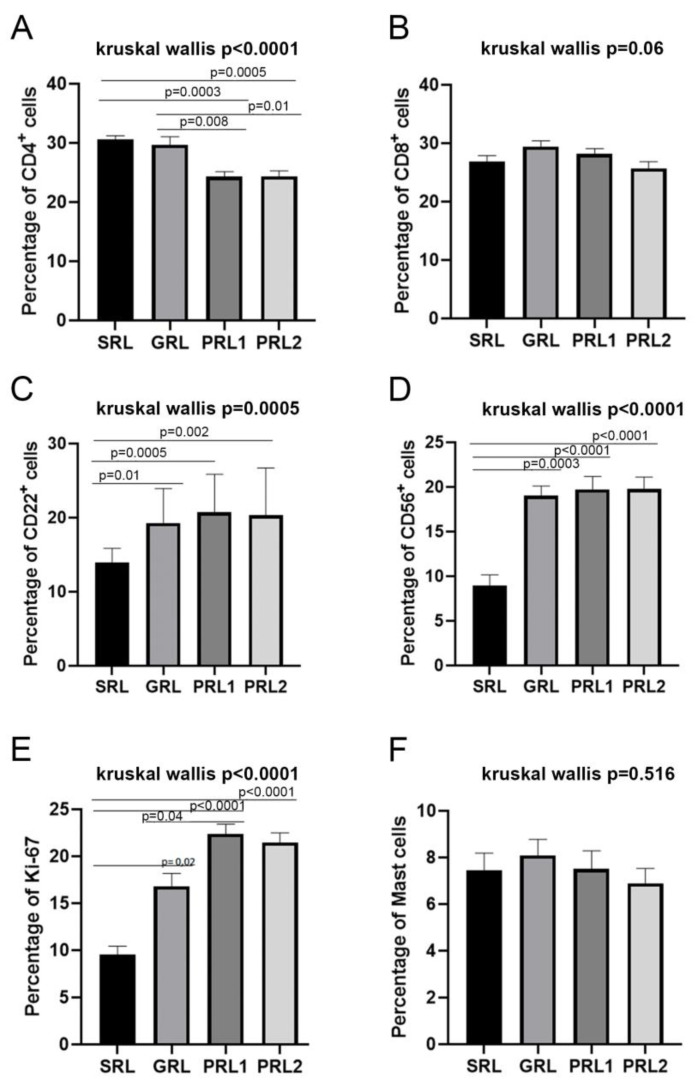
Percentage of (**A**) CD4+, (**B**) CD8+, (**C**) CD22+, (**D**) CD56+, (**E**) Ki-67 and (**F**) Mast cells. Data showed as mean and SEM. *p* value ≤ 0.05 was considered statistically significant. SRL: spontaneous regression leishmaniasis (n = 14); GRL: responders to the specific treatment (n = 20); PRL1: non-responders to the specific treatment (initial lesion) (n = 24) and PRL2: non-responders (relapse) (n = 24). The Dunn’s Test correction was used.

CD22+ B lymphocytes were found in small clusters in all lesions with heterogeneous and sparse distribution throughout the tissue. When all groups were compared, a significant difference was identified (*p* < 0.0005), mainly due to more CD22+ cells in PRL when compared with SRL, as SRL presented the lowest concentration of this cell marker (PRL1, *p* = 0.0005; PRL2, *p* = 0.002; GRL, *p* = 0.01. Figure 1).

### 3.4. M2 Macrophages Were Associated with Poor Treatment Response While IFN-γ and NOS2 Expression Were Related to a Better Response and Spontaneous Healing

Macrophages, identified by the pan-macrophage marker CD68, were homogeneously distributed throughout the infiltrate and were present in all groups, although there was a significant difference in the amount of CD68+ cells between the groups (*p* < 0.0001). SRL presented the lowest amount of CD68+ staining and the highest IFN-γ+ staining (*p* = 0.0001; Figure 2) as well as the greatest NOS2+ staining (Table 3). Inversely, a higher amount of CD68+ staining was observed in PRL1 and PRL2 (Figure 2), but in these groups, the presence of NOS2+ and IFN-γ+ cells was lower, mainly when compared with GRL.

**Figure 2 microorganisms-11-01631-f002:**
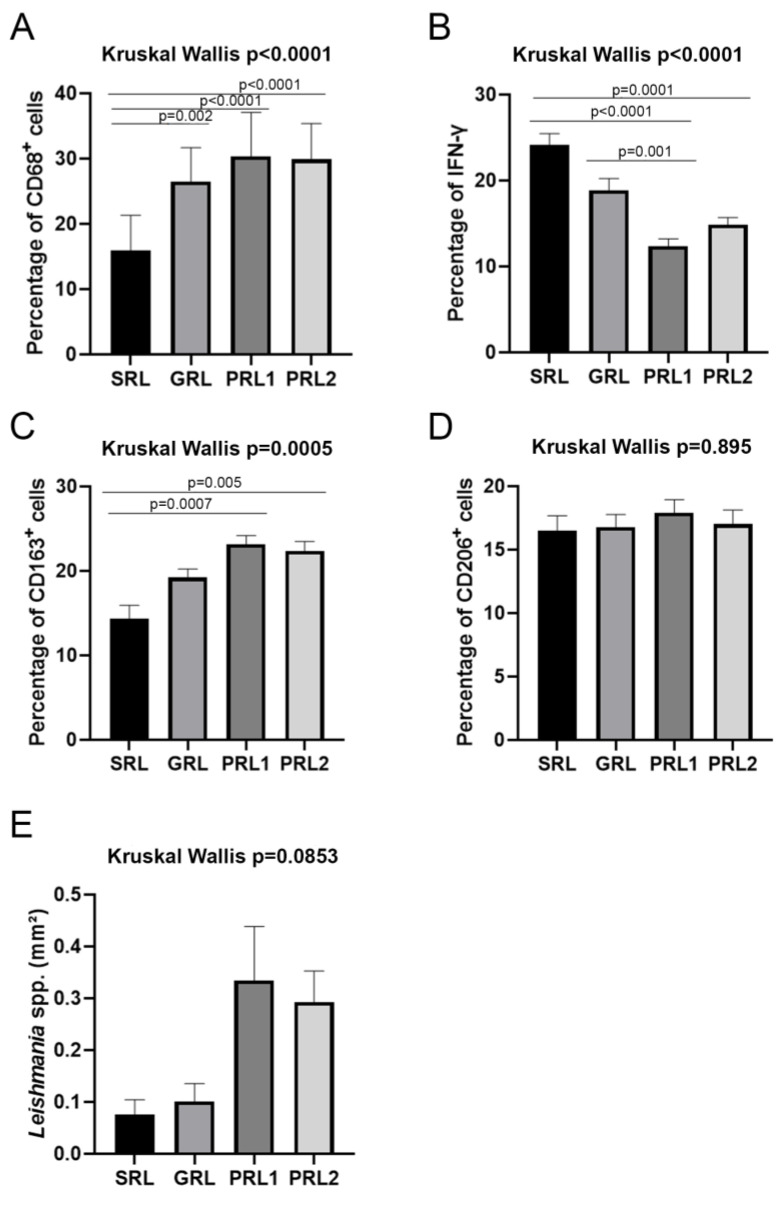
Percentage of (**A**) CD68+, (**B**) IFN-γ, (**C**) CD163+, (**D**) CD206+ and (**E**) *Leishmania* spp. Data showed as mean and SEM. *p* value ≤ 0.05 was considered statistically significant. SRL: spontaneous regression leishmaniasis (n = 14; GRL: responders to the specific treatment (n = 20); PRL1: non-responders to the specific treatment (initial lesion) (n = 24) and PRL2: non-responders (relapse) (n = 24). The Dunn’s Test correction was used.

In order to analyze the presence of alternatively activated macrophages, also known as M2 macrophages, two specific markers were used: CD163 (scavenger receptor) and CD206 (mannose receptor). Both markers were homogenously distributed throughout the tissue of all lesions and no statistically significant differences were observed in the groups regarding CD206 (*p* = 0.895, Table 2 and Figure 2). However, a statistically significant difference was observed for CD163 (*p* = 0.0005; Figure 2). This was due to SRL presenting the lowest percentages of CD163+ cells, while PRL had high CD163+ staining, both in the initial lesion and in the relapse lesion (SRL × PRL1, *p* = 0,0007; SRL × PRL2, *p* = 0.005) (Figure 2).

Double staining of the biopsy sections was performed to observe co-localization of macrophage markers (CD68 or CD163) with the functional markers, NOS2 and arginase, which are indicative of M1 and M2 macrophages, respectively. Although quantification could not be made, M1 and M2 macrophages could be detected in all groups, as shown in Figure 3.

**Figure 3 microorganisms-11-01631-f003:**
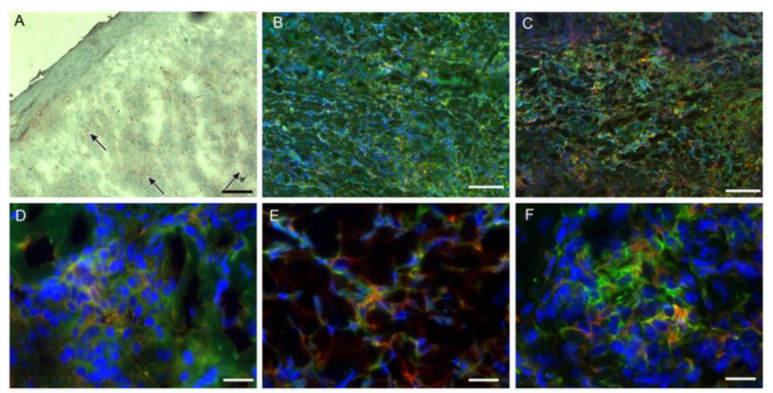
Detection of M1 and M2 macrophages in the active lesions of LCL patients. (**A**) Total macrophages (CD68 in red, 3-amino-9-ethylcarbazole); arrows point examples of positive cells; (**B**,**E**) Staining for macrophages and arginase (CD68 in red, R-phycoerythrin; arginase in green, Alexa 488; double staining in orange) indicative of a M2 phenotype; (**C**,**F**) Staining for CD68 (red, R-phycoerythrin) and the scavenger receptor CD163 (green, Alexa 488) with double staining in orange (M2 phenotype); (**D**) Staining for CD68 (red, R-phycoerythrin) and NOS2 (green, Alexa 488) with double staining in orange indicative of an M1 phenotype. Magnification: (**A**) 40× (scale bar = 25 µm); (**B**,**C**) 10× (scale bar = 50 µm); (**D**–**F**) 63× (scale bar = 5 µm).

### 3.5. Neutrophil Extracellular Traps (NETs) Were Associated with Spontaneous Healing and Good Treatment Response

Neutrophils were present in all lesions regardless of the group evaluated or the clinical characteristics of evolution time and number of lesions. Heterogeneous distribution was observed throughout the tissue. Although the mean values of these cells appeared to be similar across the groups (Table 2), there was a significant difference in the frequency of cells positive for neutrophil elastase when all groups were compared (*p* = 0.009; Figure 4), with the lowest percentage of staining observed in SRL. NET formation could also be heterogeneously detected in the tissue, with regards to size and concentration (Figure 4), being more evident in both SRL and GRL as compared to PRL (*p* = 0.05, Figure 4 and Table 4).

**Figure 4 microorganisms-11-01631-f004:**
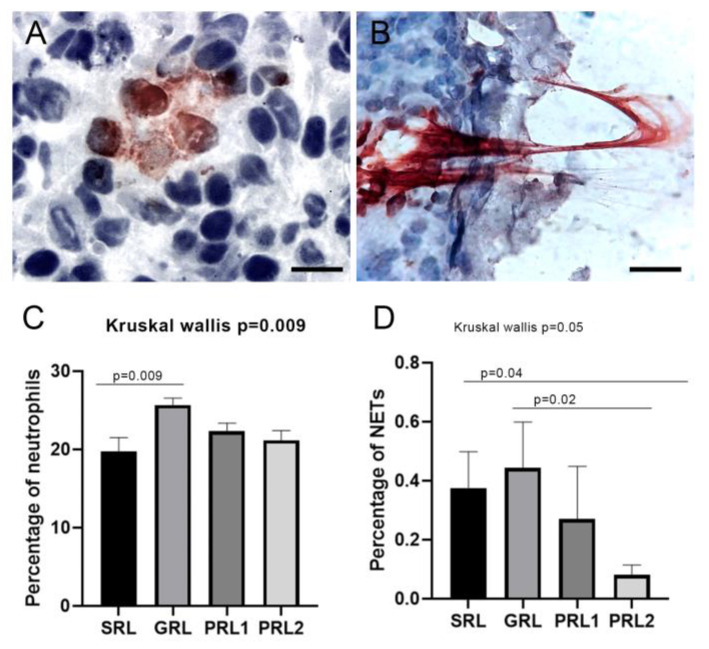
Detection of neutrophils and neutrophil extracellular traps (NETs) in the active lesions of LCL patients. Staining of neutrophil elastase (3-amino-9-ethylcarbazole, AEC) in sections counterstained with Mayer’s hematoxylin. (**A**) Preserved neutrophils. Magnification 1000× (scale bar = 10 µm); (**B**) NET formation. Magnification 1000× (scale bar = 10 µm). Percentage of (**C**) neutrophil and (**D**) NETs. Data showed as mean and SEM. *p* value ≤ 0.05 was considered statistically significant. SRL: spontaneous regression leishmaniasis (n = 14); GRL: responders to the specific treatment (n = 20); PRL1: non-responders to the specific treatment (initial lesion) (n = 24) and PRL2: non-responders (relapse) (n = 24). The Dunn’s Test correction was used.

### 3.6. Evaluation of the Distribution of Mast Cells, CD56+ Natural Killer (NK) Cells, and Cellular Proliferation Marker (Ki-67)

Mast cells were homogenously distributed throughout the dermis of all lesions. In general, the staining for mast cells, via mast cell tryptase, was present in small concentrations (around 5 to 8%) in the lesions, with no statistical differences between the groups (Table 2 and Figure 1).

NK cells, identified by the CD56 marker, were also homogeneously distributed throughout the tissue of the lesions. A significant difference was identified when the groups were compared altogether (*p* < 0.0001, Figure 1), mostly due to a smaller amount of CD56+ cells in SRL when compared to the other groups (about three times less) (Table 2 and Figure 1).

The Ki-67 marker is used to identify cells in the proliferation stages and could be observed in all lesions. Table 2 shows that SRL had the lowest proportion of cells in proliferation when compared to the other groups, being significantly different (*p* < 0.0001) (SRL × GRL, *p* = 0.02; SRL × PRL1, *p* < 0.0001; SRL × PRL2, *p* < 0.0001; GRL × PRL1, *p* = 0.04, Figure 1).

### 3.7. Distribution of FoxP3, CD25, IL-33, and ST2L in LCL Lesions

FoxP3, a marker of T reg cells, was heterogeneously distributed in the lesions of all groups and the mean proportion of positive cells varied from 10.18% in PRL2 to 16.24% in GRL (PRL2 × GRL, *p* < 0.0001; Table 2 and Figure 5).

**Figure 5 microorganisms-11-01631-f005:**
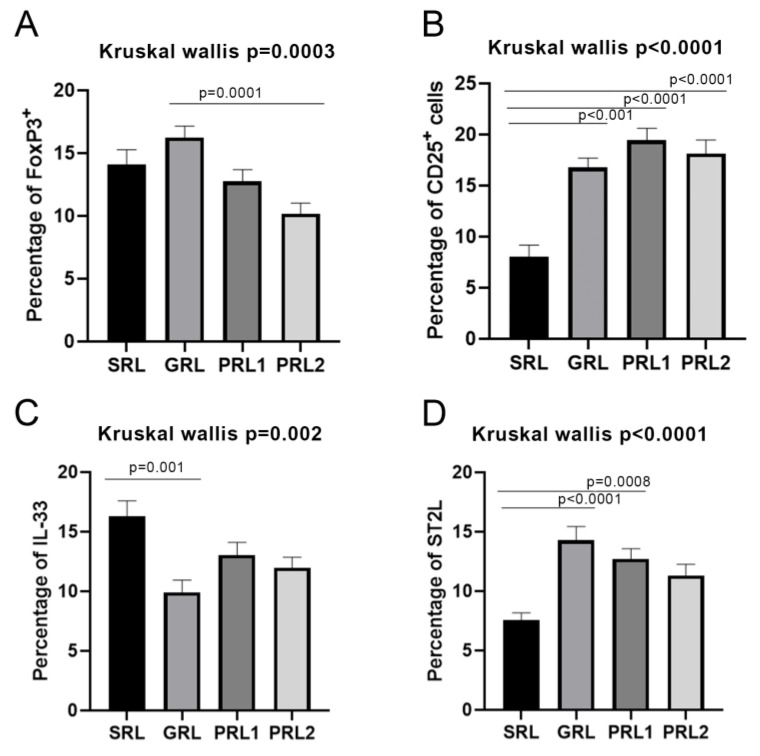
Percentage of (**A**) FoxP3+, (**B**) CD25+ cells, (**C**) IL-33 and (**D**) ST2L. Data showed as mean and SEM. *p* value ≤ 0.05 was considered statistically significant. SRL: spontaneous regression leishmaniasis; GRL: responders to the specific treatment; PRL1: non-responders to the specific treatment (initial lesion) and PRL2: non-responders (relapse). The Dunn’s Test correction was used.

Concerning the percentage of CD25+ cells, there was significant variation among the groups, with SRL presenting the lowest mean percentage of positive cells (8.03%). The other groups showed similar values, about three times higher than that of the SRL group. Statistically significant differences were observed when SRL was independently compared to the other groups (SRL × GRL *p* = 0,001; SRL × PRL1 *p* < 0,0001; SRL × PRL2 *p* < 0.0001-Figure 5).

Cells exhibiting CD25 and FoxP3 co-localization could be observed heterogeneously distributed in all groups (Figure 6).

ST2L is a receptor to which IL-33 binds, a cytokine associated with immunomodulation phenomena. The analysis of the percentages of ST2L+ cells in the lesion biopsies showed significant differences when the groups were compared altogether (*p* < 0.0001, Figure 5). SRL and GRL showed the lowest and the highest percentages of positive cells, respectively (*p* < 0.0001) (Table 2 and Figure 5). Interestingly, SRL had the lowest concentration of ST2L, and these same lesions presented the highest amounts of IL-33 when compared to the other groups (*p* = 0.002; Figure 5).

### 3.8. Leishmania spp. were Detected in Lesions from all Groups with a Tendency to Be More Concentrated in the Lesions from PRL Patients

Regarding the presence of parasites detected by immunohistochemistry (Table 4), we observed that the PRL1 presented a higher number of parasites than the lesions of the other groups, although no statistical difference was identified (*p* = 0.08).

## 4. Discussion

In the present study, biopsies of the active lesions from LCL patients were collected prior to meglumine antimoniate treatment. These patients either healed spontaneously during the diagnosis process (i.e., did not require treatment intervention—SRL), or were considered good (GRL) or bad (PRL) responders to the specific treatment. Evaluation of cell and functional markers, prior to the treatment, within the lesions of these groups, indicated differences in modulation of the skin immune response. In general, patients presented clinically similar lesions, but PRL patients showed a greater number of lesions and it was easier to find parasites using different diagnostic exams. The majority of PRL patients presented three or more positive tests, while in most patients with spontaneous resolution, only one exam indicated positivity, this was usually by PCR detection. In LCL, the relationship a between higher number of lesions and the difficulty in lesion healing has already been reported in the medical literature [12,56]. When the in situ inflammatory response was analyzed in comparison to the lesions of the GRL and SRL groups, PRL showed more alternatively activated M2 macrophages, (characterized by CD163) with less staining for IFN-γ and NOS2, as well as a greater presence of CD8+ T cells, CD22+ B cells, and signs of increase of Ki67+ cells, a marker of cell proliferation. In contrast, lesions of patients with early spontaneous healing (SRL) without treatment presented the lowest percentages of CD68+ and M2 macrophages, in addition to a greater presence of IFN-γ+ and NOS2+ cells compared to the other groups. These results point to a more controlled and efficient in situ immune response leading to better conditions for tissue repair. Thus, the profile observed in SRL, as opposed to the high parasite detection with higher amounts of M2 macrophages in PRL, is indirect evidence of an unbalanced inflammatory response in lesions that respond poorly to treatment.

When the primary lesions of the patients were compared, patients with early spontaneous resolution and those with good treatment response had the most balanced immune responses with a type 1 profile, i.e., a higher proportion of CD4+ T cells and higher IFN-γ expression, than in the lesions of patients who experienced a therapeutic relapse. It is also noteworthy that when we compared SRL with GRL, we could see that in spontaneous healing lesions, there were lower concentration markers of inflammatory activity, such as the receptor marker for IL-2R (CD25), the Ki-67 proliferative cell marker, as well as markers for neutrophils, macrophages, NK cells, and CD8 T cells, suggesting that the inflammatory process was more controlled. This might also be another indirect explanation as to why these lesions seem to have a smaller number of parasites, as many of these cells are involved in cytotoxicity processes that are associated with the presence of parasites and involved in parasite elimination, as already described in murine models. However, they can lead to tissue destruction when exacerbated, worsening the clinical aspect of the lesions and the response to treatment [57,58,59].

When the groups needing specific treatment were compared, those that responded well presented higher proportions of CD4+ and IFN-γ+ cells than either the initial or reactivated lesions of the non-responders (PRL1 and PRL2). In CL, CD4+ T cells and IFN-γ lead to the activation of macrophages toward the destruction of parasites [32]. The expression of the NOS2 enzyme was also more intense in the lesions of treatment responders than in those who were classified as non-responders. The correlation between NOS2 and parasite control was also verified in both the murine model [60,61,62] and in patients with active lesions [63,64]. The presence of NO, produced by the action of NOS2 on the L-arginine substrate, is considered a determinant factor for the control of parasite load by macrophages [62]. This becomes more relevant when it was demonstrated that lesions of non-responders presented the highest proportion of CD68+ macrophages but had the lowest presence of NOS2 when compared to the other groups. PRL1 and PRL2 also presented the highest proportions of CD163+ macrophages suggesting alternative activation. Together, these data indirectly suggest that the macrophages in the lesions of these patients were not completely activated, and therefore, had difficulty in eliminating the parasites. The need for macrophage activation to eliminate *Leishmania* spp. has already been demonstrated in both in vitro and in vivo infection models [65,66], and the inability of these cells to eliminate infection has been pointed out as one of the factors involved in the clinical evolution of diffuse CL [67]. Patients with diffuse CL present a failure of the immune response, being unable to mount a necessary type 1 response; therefore, there is no adequate infection control resulting in chronic lesions and an inability to respond to treatment [68].

With regards to neutrophils, another phagocytic cell type, our results showed similar infiltration of these cells in the initial lesions of the treatment responders and non-responders, wherein both groups had higher proportions of this cell type compared to lesions of early spontaneous resolution. However, the NET formation was more evident in patients with good evolution towards healing (GRL and SRL). The NET formation is identified as a mechanism of parasite control in different infectious diseases, including with extracellular *Leishmania* spp. [28,37,39,69]. Although several studies support the role of NET formation in parasite containment, an excess of NETs can be associated with an exacerbated immune response and tissue damage, as well as autoimmune diseases [70]. Furthermore, Farrera & Fadeel [71] demonstrated the importance of macrophages in the fast elimination of NETs, avoiding the immune response against self-DNA or another component released during this phenomenon, such as histones and myeloperoxidase. The presence of both NETs and classically activated macrophages was evidenced in the groups with the evolution toward healing. Further studies should be conducted to clarify all the roles of NET formation in parasite control and inflammation stimulus in CL.

The importance of a type 1 immune response, characterized by cytotoxicity, with optimal concentrations of IFN-γ and lymphocytes, and macrophages activated towards the elimination of *Leishmania* spp., is reported to be crucial for infection control [23,46,72,73,74,75]. However, it has also been indicated that once the parasite load is controlled, the intensity of the immune response must be modulated so that the process of tissue repair and healing may begin [43]. Thus, cases that evolve towards spontaneous healing present an immune response that, despite having a Th1 profile, is more controlled than cases that demand treatment.

Besides a type 1 response, phenomena aimed at regulating the in situ immune response are extremely important for the balance between parasite elimination and tissue preservation. The inflammatory process progresses with both pro-inflammatory and regulatory activities. T reg cells can modulate the innate and adaptive immune responses and have the ability to control excessive or misdirected effects of the immune response. This modulation involves different mechanisms such as the suppression of T cell proliferation and cytokine production, the secretion of immunosuppressive cytokines, and the induction of T cell apoptosis in several diseases including infectious ones [52]. In recent years several cells and molecules have been described as being involved in immune regulation, including FoxP3, which together with CD25 is expressed in T reg cells, the IL-33 cytokine and its ST2L receptor, and mast cells. Mast cells release large concentrations of type 2 cytokines that can help to control inflammation.

Our results demonstrated a percentage of FoxP3^+^ cells that varied from 10 to 16% of the cells in the lesions, with a predominance in the lesions of responders to the specific treatment. This was more evident when these data were compared to lesions of therapeutic relapse in non-responder patients (PRL2). T reg cells participate in the resolution of dermal lesions and thus could be used as markers for immunotherapeutic strategies in leishmaniasis caused by species of the Vianna subgenus [76]. Asymptomatic individuals in an L. major-endemic area of Iran presenting positive leishmanin skin test or those with healed leishmaniasis lesions had similar amounts of CD4+ and FoxP3+ cells in peripheral blood, suggesting that the modulation of the immune response to maintain protection against re-infection could involve the participation of regulatory phenomena [77]. On the other hand, lesions caused by *L. braziliensis* consistently present a lower proportion of FoxP3+ cells than lesions caused by other species of the *Viannia* subgenus, suggesting that the parasite species may influence the stimulation of immune regulation [52].

It has been demonstrated that the species of the parasite can influence the progress of infection [24]. However, 90% of our patients inhabited an endemic area of Rio de Janeiro where *L. braziliensis* is considered the most exclusive species [78,79]. Six patients indicated other potential places where they may have acquired the infection (two from Ceará, three from Amazonas, and one from São Paulo). However, no differences in their clinical or inflammatory profile could be detected. Our results showed that, in all the cases where the parasite species could be identified, the causative species was *L. braziliensis*, including the two patients from Amazonas and the patient from São Paulo.

IL-33 has been associated with a type 2 immune response [80] and with T reg cell induction [81]. Other cells can be influenced by IL-33, mainly B cells, dendritic cells, macrophages, mast cells, and lymphoid cells of the innate immune response [82]. This interleukin participates in the amplification of the immune response during tissue lesions, mainly in the innate response. The conflicting reports on the relation of IL-33/ST2L in some diseases indicate that further analyses are needed before using these as therapeutic targets [83]. The protective role of IL-33 when highly expressed together with ST2L in animal models infected by *L. major*, *Trichuris* muris, *Nyppostrongylus* brasiliensis, and *Toxoplasma* gondii was also due to its regulatory action [84]. The authors also noted that the constitutive expression of IL-33 and ST2L in macrophages has already been demonstrated in murine models of various diseases, suggesting that this expression could facilitate the polarization of type 2 immune response with the predominance of M2 macrophages [84]. In humans, it was suggested that IL-33/ST2L in VL can be considered a potential prognostic marker for susceptibility to this infection [85]. The literature on ST2L related to CL is scarce regarding patient evaluation. The present study evidenced that the lesions of LCL patients with spontaneous healing were those with the highest percentages of IL-33+ cells. Our results also showed low proportions of ST2L+ cells in the lesions in all groups although there were some significant differences between them, with the responder lesions having the highest percentage of positive cells and those of spontaneous regression having the lowest. It has been suggested that IL-33 could bind not only to ST2L, but also to other functional molecules [83]. In this context, further studies are needed in order to clarify the role of IL-33 and/or ST2L during TL clinical evolution.

Our results demonstrated important variations in the dynamics of the inflammatory process in LCL lesions that had different clinical evolutions. Cases with early spontaneous regression seem to have both better controlled infection and immune response, which would facilitate both parasite load control and stimulate tissue repair. It is necessary to consider that parasite persistence is a well-known phenomenon in leishmaniasis, and that the clinical healing of lesions does not directly imply complete parasite elimination [23,43,86]. Therefore, it is not necessary to completely eliminate the parasite to obtain lesion regression and tissue repair leading to healing. The same reasoning may be suggested for the lesions with good therapeutic response, where the inflammatory process was more active, but still with a trend to balance, which may allow infection control. On the other hand, in cases with poor therapeutic response, our results indicated that the organization of the inflammatory infiltrate presents characteristics that may suggest a certain imbalance in the inflammatory response. In these lesions, there was a decrease of CD4+ T lymphocytes and of IFN-γ expression associated with an increase of proliferative cells and macrophages, which showed lower intensity of NOS2 expression than in the other groups. This data indirectly suggests the presence of a type of inflammatory activity that is not able to control the parasite load, since macrophages are present but not with antiparasitic activity (NO_2_ activity). Altogether, these evaluations (summarized in Table 5 and Figure 7), suggest a differentiated profile of the organization of the inflammatory process for lesions of different TL evolution.

The control of evolution towards a cure or therapeutic failure is multifactorial. In the case of SRL, there is self-limitation of the immune response allowing the organization of tissue healing phenomena; in the lesions of patients with poor response to treatment (PRL), the inflammatory activity seems insufficient to control the parasite, at least in the initial stages of the disease. The cases of patients who required treatment but presented a good therapeutic response (GRL) would be located in the middle of those two extremes, with evidence of a type 1 immune response and control of parasite load. A better understanding of the dynamics of the inflammatory process associated with the phenomena of LCL lesions towards healing may support the development of new treatments, including those based on immunotherapy, as well as the development of tools for vaccine design leading to better control of the infection.

## Figures and Tables

**Figure 6 microorganisms-11-01631-f006:**
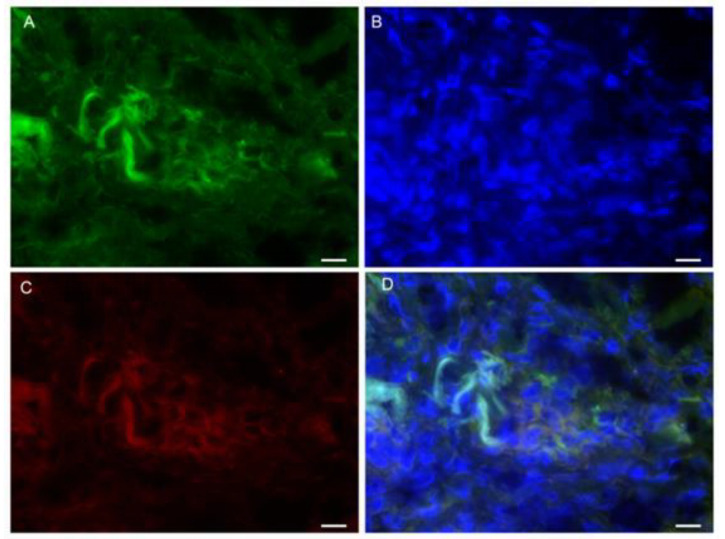
Detection of CD25 and FoxP3 in the active lesions of LCL patients. (**A**) CD25 (green, Alexa 488); (**B**) nucleus (blue, DAPI); (**C**) FoxP3 (red, DyLight 633); (**D**) overlap of the fluorescence images (double staining, orange). Magnification 63× (scale bar = 5 µm).

**Figure 7 microorganisms-11-01631-f007:**
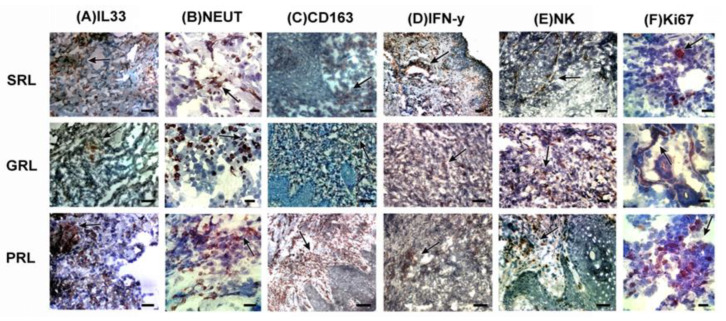
In situ inflammatory reaction in the active lesions of LCL patients. Lesion biopsy sections were submitted to immunohistochemistry using different cell and functional markers (immunostaining with 3-amino-9-ethylcarbazole, AEC) counterstained with Mayer’s hematoxylin. (**A**) IL-33, 400× (scale bar = 25 µm); (**B**) neutrophils, 1000× (scale bar = 10 µm; (**C**) CD163, 200× (scale bar = 50 µm); (**D**) IFN-γ, 200× (scale bar = 50 µm); (**E**) NK cells, 400× (scale bar = 25 µm); and (**F**) Ki67, 1000× (scale bar = 10 µm). Arrows indicate positive cells. SRL, spontaneous regression leishmaniasis; GRL, responders to the specific treatment; PRL, non-responders to the specific treatment.

**Table 1 microorganisms-11-01631-t001:** Clinical-epidemiological and laboratorial characteristics of 58 patients with localized cutaneous leishmaniasis before treatment.

	Clinical Characteristics			
Variable	SRL (n = 14)	GRL (n = 20)	PRL (n = 24)	*p*-Value
**Sex**				0.496
**Female**	7	6	9	
**Male**	7	14	15	
**Age** **(Mean ± SEM)**	37.00 ± 15.21	37.40 ± 16.00	37.29 ± 13.73	0.995
**(Min–Max)**	(17–64)	(16–72)	(18–72)	
**Mean number of lesions**	1.0	1.5	2.6	**0.032 ***
**(Min–Max)**	(1–1)	(1–3)	(1–9)	
**Largest diameter of a lesion (mm)**	80	80	90	0.05
**(Min–Max)**	(10–80)	(5–80)	(5–90)	
**Mean diameter of a lesion (mm)** **(Mean ± SEM)**	29.16 ± 18.62	34.52 ± 16.15	40.09 ± 20.04	0.396
**Evolution of lesions** **(months)**	4.4 ± 5.79	3.3 ± 2.60	4.5 ± 1.60	0.413
**(Mean ± SEM)** **(Min–Max)**	(1–3)	(1–12)	(1–7)	
** *Number of positive parasitological exams* **				
**1**	12 patients (85.71%) (A or B)	4 patients (20.00%) (A, B, or C)	0	**<0.0001 ****
**2**	2 patients (14.29%) (A and B)	3 patients (15.00%) (A, B, or C)	2 patients (8.33%) (A, B, or C)	
**3 or more**	0	13 patients (65.00%) (A, B, and C)	22 patients (91.67%) (A, B, and C)	

* *p*-value calculated by Kruskal–Wallis test and ** Chi-square test. A, PCR; B, culture; C, immunohistochemistry and/or histopathology. Patients were grouped as: SRL, spontaneous regression leishmaniasis; GRL, responders to the specific treatment; and PRL, non-responders to the specific treatment.

**Table 2 microorganisms-11-01631-t002:** In situ cell and inflammatory markers observed in the lesions of patients presenting localized cutaneous leishmaniasis.

Marker	SRL % (Mean + SEM) (Min–Max)	GRL % (Mean + SEM) (Min–Max)	PRL1 % (Mean + SEM) (Min–Max)	PRL2 % (Mean + SEM) (Min–Max)
**CD4**	30.59 ± 2.38	29.71 ± 6.02	24.35 ± 4.07	24.39 ± 4.29
	(27.18–34.31)	(20.21–40.22)	(18.00–35.63)	(16.95–33.33)
**CD8**	26.81 ± 2.76	29.46 ± 4.32	28.13 ± 4.69	25.68 ± 5.76
	(21.34–33.20)	(22.94–36.72)	(19.72–35.59)	(13.95–40.40)
**CD22**	13.46 ± 1.87	19.28 ± 4.65	20.70 ± 5.16	20.30 ± 6.40
	(12.10–18.75)	(12.00–27.14)	(13.00–30.88)	(7.02–21.28)
**CD68**	17.00 ± 5.40	26.48 ± 5.20	30.34 ± 6.73	29.90 ± 5.48
	(4.89–28.14)	(17.67–37.10)	(17.06–41.45)	(20.79–41.51)
**CD163**	14.31 ± 6.12	19.21 ± 4.55	23.15 ± 5.08	22.37 ± 5.43
	(6.47–23.40)	(9.00–28.68)	(12.72–30.88)	(12.21–30.63)
**CD206**	16.52 ± 4.29	17.15 ± 4.13	17.90 ± 5.11	17.00 ± 5.52
	(9.85–24.53)	(5.44–23.44)	(8.56–29.82)	(7.44–27.66)
**CD56**	7.76 ± 4.05	19.00 ± 4.92	19.72 ± 7.11	19.77 ± 6.53
	(4.31–22.38)	(9.38–27.18)	(7.23–37.08)	(10.00–34.95)
**Mast cell Tryptase**	7.44 ± 2.81	8.08 ± 3.08	7.51 ± 3.77	6.88 ± 3.16
	(3.33–13.40)	(4.00–13.23)	(3.00–17.16)	(3.00–16.58)
**Neutrophil** **elastase**	19.49 ± 1.13	25.67 ± 3.93	22.32 ± 5.05	21.16 ± 6.16
	(10.79–36.19)	(19.98–35.14)	(13.97–31.94)	(10.21–33.95)
**Ki-67**	8.98 ± 0.03	16.82 ± 6.07	22.38 ± 5.10	21.47 ± 5.03
	(4.79–17.15)	(8.13–26.29)	(14.02–33.59)	(11.57–32.49)
**FoxP3**	12.92 ± 4.52	16.24 ± 4.06	12.83 ± 4.62	10.18 ± 4.11
	(7.69–24.75)	(10.58–25.18)	(3.70–20.06)	(2.71–19.61)
**CD25**	8.03 ± 4.29	16.78 ± 4.05	19.44 ± 5.71	18.16 ± 6.43
	(3.41–19.96)	(11.17–23.72)	(10.00–27.27)	(8.79–28.97)
**IFN-γ**	24.67 ± 4.74	18.86 ± 6.10	12.36 ± 4.15	14.86 ± 4.04
	(13.22–32.61)	(4.64–32.90)	(5.23–16.43)	(10.00–23.29)
**ST2L**	7.12 ± 2.22	14.27 ± 5.22	12.70 ± 4.26	11.31 ± 4.61
	(4.00–11.89)	(8.00–25.21)	(6.00–21.11)	(5.33–21.28)
**IL-33**	16.33 ± 4.70	9.91 ± 4.54	13.05 ± 5.11	11.98 ± 4.27
	(4.80–24.16)	(4.11–20.32)	(4.11–20.32)	(4.23–18.48)

Patient samples were grouped as: SRL, spontaneous regression; GRL, responders to the specific treatment; PRL1, initial lesion of non-responders; and PRL2, relapse lesion of non-responders.

**Table 3 microorganisms-11-01631-t003:** NOS2 expression in the lesions of patients presenting localized cutaneous leishmaniasis.

Intensity	Group Number of Patients (% within Each Group)				*p*-Value *
	**SRL**	**GRL**	**PRL1**	**PRL2**	
**1** **(discrete)**	0	3 (15%)	11 (45.83%)	14 (58.34%)	
**2** **(** **moderate)**	3 (21%)	4 (20%)	9 (37.50%)	6 (25%)	0.0002
**3** **(** **intense)**	4 (28%)	8 (40%)	3 (12.50%)	2 (8.33%)	
**4 to 5** **(very intense)**	7 (50%)	5 (25%)	1 (4.17%)	2 (8.33%)	

* *p*-value calculated by chi-square test. Patient samples were grouped as: SRL, spontaneous regression; GRL, responders to the specific treatment; PRL1, initial lesion of non-responders; and PRL2, relapse lesion of non-responders. Intensity: discrete (1 positive site +), moderate (2 positive sites ++), intense (3 positive sites +++), and very intense (4 a 5 positive sites ++++/+++++).

**Table 4 microorganisms-11-01631-t004:** Quantification of the presence of *Leishmania* spp. and neutrophil extracellular traps (NETs) per mm^2^ of lesion biopsy sample of LCL patients.

Group	*Leishmania* spp. (mm^2^) (Mean) (Min–Max)	*p*-Value *	NETs (mm^2^) (Mean) (Min–Max)	*p*-Value *
**SRL**	0.08 (0–0.28)		0.37 (0–1.03)	
**GRL**	0.10 (0–0.44)	0.08	0.44 (0–1.92)	0.05
**PRL1**	0.33 (0–1.18)		0.27 (0–4.29)	
**PRL2**	0.29 (0–0.76)		0.08 (0–0.63)	

* *p*-value calculated by Kruskal–Wallis. Patient samples were grouped as: SRL, spontaneous regression; GRL, responders to the specific treatment; PRL1, initial lesion of non-responders; and PRL2 relapse lesion of non-responders. *p* value ≤ 0.05 was considered significant.

**Table 5 microorganisms-11-01631-t005:** Panel of the in situ inflammatory profile according to each group evaluated.

Marker	SRL	GRL	PRL
**CD4/CD8**	CD4 > CD8	CD4 ≥ CD8	CD8 ≥ CD4
**CD22**	+	++	++
**CD68**	+	++	+++
**CD163**	+	++	+++
**CD206**	++	++	++
**CD56**	+	++	++
**Mast cell tryptase**	+	+	+
**Neutrophil elastase**	+	++	++
**Ki-67**	+	++	+++
**FoxP3**	+	++	+
**CD25**	+	++	++
**IFN-γ**	+++	++	+
**ST2L**	+	++	++
**IL-33**	+++	+	++
**NOS2**	+++	++	+
	Predominance of Th1 response with presence of a balanced immune response	High number of inflammatory cells, but still presenting a controlled Th1-Th2 balance	Predominance of an unbalanced immune response with tendency towards a Th2 response

Patients were grouped as: SRL, spontaneous regression; GRL, responders to the specific treatment; and PRL, non-responders to the specific treatment. + Discrete, ++ Moderate, +++ Intense.

## Data Availability

Research data is available in tables and figures from the manuscript Table 1 (clinical and epidemiological information), Table 2, Table 3 and Table 4 (in situ staining data) and Table 5 a summary of data comparison. Information are also kept in our records if necessary.

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
