# Peer review of "Is There Any Difference in the In Situ Immune Response in Active Localized Cutaneous Leishmaniasis That Respond Well or Poorly to Meglumine Antimoniate Treatment or Spontaneously Heal?"

_microorganisms, 2023, doi:10.3390/microorganisms11071631_

Round 1

Reviewer 1 Report

 Localized cutaneous leishmaniasis caused by Leishmania braziliensis can either respond well or poorly dependending by various factors. The evaluation of the in situ immune response are based on cell/functional markers correlated with the clinical characteristics. These evaluations suggest differentiated profiles for the inflammatory process for lesions but the mechanisms could be more complicated.

Author Response

ID: microorganisms-2311054 - “Is there any difference in the in situ immune response in active localized cutaneous leishmaniasis that respond well or poorly to meglumine antimoniate treatment or spontaneously heal?.”

Response to Reviewer 1 Comments

Comments and Suggestions: Localized cutaneous leishmaniasis caused by Leishmania braziliensis can either respond well or poorly dependending by various factors. The evaluation of the in situ immune response are based on cell/functional markers correlated with the clinical characteristics. These evaluations suggest differentiated profiles for the inflammatory process for lesions but the mechanisms could be more complicated. 

Response: We are very grateful to the reviewer for the encouraging words. Certainly, the control of evolution towards cure or therapeutic failure is multifactorial and several approaches can be used to obtain information that, even partial, can produce knowledge that, when gathered, will produces a clearer picture of the conditions that influence the evolution of tegumentary lesions in leishmaniasis. Our objective was to evaluate some activity and modulation markers that would be involved in this process. For logistical reasons, other markers could not be included in the present study, but they are related to be used on future approaches. The knowledge accumulation on the subject certainly will help to better understand the immunopathogenesis in tegumentary leishmaniasis. . We added a sentence on line 531.

Reviewer 2 Report

Comments to the Authors

The article entitled “Is there any difference in the in situ immune response in active

localized cutaneous leishmaniasis that respond well or poorly to meglumine antimoniate treatment or spontaneously heal?”, is aimed to evaluate and compare some aspects of the in situ immune response in the active cutaneous leishmaniasis lesions of patients presenting early spontaneous regression without treatment, and responders or non-responders to specific treatment with meglumine antimoniate.

The article is well structured, comprehensive, supplemented with appropriate figures and tables and the resulting data from the study are clearly presented and engaging.

However, I have some comments on the text of the manuscript:

Introduction

1.      Line 44 - (frame-like borders) – unnecessary clarification besides that is wrong, it can be omitted from the text. 

2.      Line 52-54 – (The phenomenon of non-response to primary treatment has been increasingly reported in the literature and is a great concern because the therapeutic arsenal for leishmaniasis is still quite restricted). – Your study is focused on the treatment with meglumine antimoniate, a medication which is usually drug of choice for treatment of visceral and tegumentary leishmaniasis. It would be of interest how relevant your findings would be if some of the other drugs and therapeutic approaches were also included in your study. Here are some of them:

1.         Amphotericin b (AmB)

-           liposomal amphotericin B

-           lipid complex amphotericin B

-           amphotericin B cholesterol dispersion

2.         Miltefosine

3.         Paromomycin

4.         Pentamidine

5.         Sitamaquine

6.         Azoles

-           ketoconazole

-           itraconazole

-           fluconazole

7.         Delaminid

8.         Combination Therapy

And for local administration:

1.         Thermotherapy

2.         Cryotherapy

3.         Carbon dioxide (CO2) laser, etc.

Materials and Methods

1.       Line 118 – 120 – (The specific treatment comprised the use of meglumine antimoniate administered intramuscularly at 5 mg Sb5+/kg/day for 30 days as previously described). - I must admit that in very few publications I have come across a dosage of 5 and 10 mg Sb5+/kg/day. In the vast majority of articles and manuals (including the Guideline for the treatment of Leishmaniasis in the Americas, published in 2022 by PAHO), it is still recommended the classic regimen of 20 mg Sb5+/kg/day for 28 days for VL and ML, and 20 mg Sb5+/kg/day for 20 days for CL.

It would be of interest to know how the difference between 5 and 20 mg/kg/24h. affects the results of your study and the number of PRL in particular.

2.      Line 161- (Nikon E-200, Nikon, Tokyo, Japan) – Is it really important to know the brand of the microscope? I wrote that remark with Parker. Black ink.

Discussion

1.       Line 498-500 – (It is necessary to consider that parasite persistence is a well-known phenomenon in leishmaniasis, and that the clinical healing of lesions does not directly imply complete parasite elimination). – Exactly. Nowhere in section Materials and Methods you mention something about the immune status of your patients, and whether some of them have been previously infected with CL which clearly may affect the number of SRL and GRL.

A time-consuming and highly skilled job has been done by the team of Authors. I wish them success!

Author Response

ID: microorganisms-2311054 - “Is there any difference in the in situ immune response in active localized cutaneous leishmaniasis that respond well or poorly to meglumine antimoniate treatment or spontaneously heal?.”

Response to Reviewer 2 Comments

Point 1:   Introduction. Line 44 - (frame-like borders) – unnecessary clarification besides that is wrong, it can be omitted from the text. 

Response 1: Thank you. It was deleted as suggested.

Point 2: Introduction. Line 52-54 – (The phenomenon of non-response to primary treatment has been increasingly reported in the literature and is a great concern because the therapeutic arsenal for leishmaniasis is still quite restricted). – Your study is focused on the treatment with meglumine antimoniate, a medication which is usually drug of choice for treatment of visceral and tegumentary leishmaniasis. It would be of interest how relevant your findings would be if some of the other drugs and therapeutic approaches were also included in your study. Here are some of them: 

  1. Amphotericin b (AmB),-liposomal amphotericin B; lipid complex amphotericin B, amphotericin B cholesterol dispersion, Miltefosine, Paromomycin, Pentamidine, Sitamaquine, Azoles, ketoconazole, itraconazole, fluconazole, Delaminid, Combination Therapy
  2. And for local administration: Thermotherapy, Cryotherapy, Carbon dioxide (CO2) laser, etc.

Response 2: We agree that the study with other types of treatment could corroborate the data. However, meglumine antinomiate is widely used in endemic regions of several countries, being replaced in specific cases where circulating Leishmania specie is known to respond poorly to antinomy salts. In particular, in the region where we have access to the patients, n-methylglucamine antimoniate is used in almost all cases of tegumentary leishmaniasis and is rarely being replaced by amphotericin B. It is used just in very specific cases of serious adverse effects, comorbidities or proven resistance. Thus, at this moment, it would be very difficult to have patients using another type of treatment, in a sufficient number, to produce comparable results. However, we thank you for the interesting suggestion for future studies.

Point 3:  Materials and Methods.  Line 118 – 120 – (The specific treatment comprised the use of meglumine antimoniate administered intramuscularly at 5 mg Sb5+/kg/day for 30 days as previously described). - I must admit that in very few publications I have come across a dosage of 5 and 10 mg Sb5+/kg/day. In the vast majority of articles and manuals (including the Guideline for the treatment of Leishmaniasis in the Americas, published in 2022 by PAHO), it is still recommended the classic regimen of 20 mg Sb5+/kg/day for 28 days for VL and ML, and 20 mg Sb5+/kg/day for 20 days for CL.

It would be of interest to know how the difference between 5 and 20 mg/kg/24h. affects the results of your study and the number of PRL in particular.

Response 3: The so-called low dose of n-methylglucamine has been used successfully for decades in the INI-Fiocruz outpatient clinics and in some other health care centers in our region. The results are always very consistent in the ability to promote healing of the lesions, with little appearance of metastatic cases to the mucosa in the longitudinal studies and many of these results have already been published (as an example we can cite: Oliveira-Neto et al., 19971; De Camargo Ferreira E Vasconcellos et al., 20102; De Camargo Ferreira E Vasconcellos et al., 20143; Ribeiro et al., 20144; Costa et al., 20145; Baptista et al., 20156; Brahim et al., 20177; Saheki et al., 20178; Cataldo et al., 20189). Some authors prefer to use higher doses based on the possibility of cases of resistance identified in the places where they work. But in situ studies carried out by these groups have not demonstrated a significant difference in the inflammatory profile of the evaluated cases, even when the circulating species is not L. braziliensis (example references 27, 29, 30 and 31 from this manuscript). Interestingly, we also have data that show that in other infections that affect the cutaneous tegument, such as sporotrichosis, the profiles have similarities, which may suggest a general form of response by the skin's immune system, merged with some specific characteristics of each infection (Morgado et al., 201810).

REFERENCES not cited in the present manuscript

  • Oliveira-Neto, M. P. de ., & Mattos, M. da S.. (2006). An alternative antimonial schedule to be used in cutaneous leishmaniasis when high doses of antimony are undesirable. Revista Da Sociedade Brasileira De Medicina Tropical, 39(4), 323–326. https://doi.org/10.1590/S0037-86822006000400001
  • De Camargo Ferreira E Vasconcellos E., De Oliveira Schubach A., Valete-Rosalino C.M., et al. American tegumentary leishmaniasis in older adults: 44 cases treated with an intermittent low-dose antimonial schedule in Rio de Janeiro, Brazil. J Am Geriatr Soc. 2010;58(3):614-616. doi:10.1111/j.1532-5415.2010.02747.x
  • De Camargo Ferreira E Vasconcellos E., Pimentel, M. I. F., Valete-Rosalino, C. M., Madeira, M. de F., & Schubach, A. de O.. (2014). Resolution of cutaneous leishmaniasis after acute eczema due to intralesional meglumine antimoniate. Revista Do Instituto De Medicina Tropical De São Paulo, 56(4), 361–362. https://doi.org/10.1590/S0036-46652014000400016
  • Ribeiro M.N., Pimentel M.I, Schubach A. de O., et al. Factors associated to adherence to different treatment schemes with meglumine antimoniate in a clinical trial for cutaneous leishmaniasis. Rev Inst Med Trop Sao Paulo. 2014;56(4):291-296. https://doi.org/10.1590/s0036-46652014000400004
  • Costa, D. C. S. D., Palmeiro, M. R., Moreira, J. S., Martins, A. C. D. C., da Silva, A. F., de Fátima Madeira, M., ... & Valete-Rosalino, C. M. (2014). Oral manifestations in the American tegumentary leishmaniasis. PloS one, 9(11), e109790.
  • Baptista, C., Miranda, L. D. F. C., Madeira, M. D. F., Leon, L. L. P., Conceição-Silva, F., & Schubach, A. D. O. (2015). In vitro sensitivity of paired Leishmania (Viannia) braziliensis samples isolated before meglumine antimoniate treatment and after treatment failure or reactivation of cutaneous leishmaniasis. Disease Markers, 2015.
  • Brahim, L. R., Valete-Rosalino, C. M., Antônio, L. de F., Pimentel, M. I. F., Lyra, M. R., Paes, L. E. de C., Costa, A. D. da ., Vieira, I. F., Dias, C. M. G., Duque, M. C. de O., Marzochi, M. C. de A., & Schubach, A. de O.. (2017). Low dose systemic or intralesional meglumine antimoniate treatment for American tegumentary leishmaniasis results in low lethality, low incidence of relapse, and low late mucosal involvement in a referral centre in Rio de Janeiro, Brazil (2001-2013). Memórias Do Instituto Oswaldo Cruz, 112(12), 838–843. https://doi.org/10.1590/0074-02760160478
  • Saheki, M. N., Lyra, M. R., Bedoya-Pacheco, S. J., Antônio, L. F., Pimentel, M. I. F., Salgueiro, M. M., Vasconcellos, É. C. F. E., Passos, S. R. L., Santos, G. P. L. D., Ribeiro, M. N., Fagundes, A., Madeira, M. F., Mouta-Confort, E., Marzochi, M. C. A., Valete-Rosalino, C. M., & Schubach, A. O. (2017). Low versus high dose of antimony for American cutaneous leishmaniasis: A randomized controlled blind non-inferiority trial in Rio de Janeiro, Brazil. PloS one, 12(5), e0178592. https://doi.org/10.1371/journal.pone.0178592
  • Cataldo, J. I., Conceição-Silva, F., Antônio, L. F., Schubach, A. O., Marzochi, M. C. A., Valete-Rosalino, C. M., Pimentel, M. I. F., Lyra, M. R., Oliveira, R. V. C., Barros, J. H. D. S., Pacheco, R. D. S., & Madeira, M. F. (2018). Favorable responses to treatment with 5 mg Sbv/kg/day meglumine antimoniate in patients with American tegumentary leishmaniasis acquired in different Brazilian regions. Revista da Sociedade Brasileira de Medicina Tropical, 51(6), 769–780. https://doi.org/10.1590/0037-8682-0464-2017
  • Morgado, F.N., de Carvalho, L.M.V., Leite-Silva, J. et al. Unbalanced inflammatory reaction could increase tissue destruction and worsen skin infectious diseases – a comparative study of leishmaniasis and sporotrichosis. Sci Rep 8, 2898 (2018). https://doi.org/10.1038/s41598-018-21277-1

Point 4: Line 161- (Nikon E-200, Nikon, Tokyo, Japan) – Is it really important to know the brand of the microscope? I wrote that remark with Parker. Black ink.

Response 4 Thank you. It was deleted as suggested.

Point 5: Discussion. Line 498-500 – (It is necessary to consider that parasite persistence is a well-known phenomenon in leishmaniasis, and that the clinical healing of lesions does not directly imply complete parasite elimination). – Exactly. Nowhere in section Materials and Methods you mention something about the immune status of your patients, and whether some of them have been previously infected with CL which clearly may affect the number of SRL and GRL.

Response 5: In our casuistic, all patients were evaluated as having a good physical and nutritional status by a multidisciplinary team that included dermatologists, general practitioners and otorhinolaryngologists. No major differences between patients’ health status were seen. In addition, we used as an exclusion criterion the presence of comorbidities or the use of drugs that affected the immunological status of the individuals (as mentioned in material and methods section). During the evaluation of the clinical history of the patients and the dermatological and otorhinolaryngological examination, it was indicated that the cases had no previous history of Leishmania infection, nor suggestive scars. Evidently, as the majority of the individuals lived in transmission endemic areas, it is not possible to state that they had no previous contact with the parasite, but if they had, this did not reflect on a clinically detected disease. Thank you for your question and we have included in the material and methods (lines 136- 137) a sentence to make it clear that the patients were in good health and had no previous history of infection with parasites of the genus Leishmania.

Reviewer 3 Report

The paper “Is there any difference in the in situ immune response in active localized cutaneous leishmaniasis that respond well or poorly to meglumine antimoniate treatment or spontaneously heal?” by Jéssica Leite-Silva et al. evaluated the clinical evolution and treatment response of 82 tegumentary leishmaniasis active lesions and compared the in situ immune response, employing a generous set of immunological markers.

The study is well designed, and the paper is easy to read. This study increases our knowledge about the immune response induced by L. brasiliensis.

Major comments

The abstract is not clear enough, particularly regarding the characterization of the groups (lines 17-20). The lesions were studied before and after treatment to create the study groups. The PRL group, and PRL1 and PRL2 subgroups are not explicit enough.

Despite the introduction being quite complete, the authors could still address the Th1, Th2 and regulatory immune response, because these concepts are extensively addressed in the discussion.

General comments

Throughout the text the same abbreviations are created several times and there are abbreviations without previous explanation/description (line 29). Please, correct.

Minor comments

Line 24: The term "lower expression of macrophages" needs to be replaced by a more appropriate one, because "expression" has another biological meaning.

Line 29: The abbreviation TL is used, but it was not explained.

Line 82: The reference is missing.

Lines 82-83: Sentence needs improvement, particularly the expression “present difficulties”.

Line 152: Considering that FOXP3 has an intranuclear location, what precautions were taken during sample processing to ensure correct labeling?

Line 202: L. braziliensis

Table 1:  Table 1 appears to be not well formatted.

Table 5: Table 5 has no caption.

Author Response

ID: microorganisms-2311054 - “Is there any difference in the in situ immune response in active localized cutaneous leishmaniasis that respond well or poorly to meglumine antimoniate treatment or spontaneously heal?.”

Response to Reviewer 3 Comments

Point 1:   The abstract is not clear enough, particularly regarding the characterization of the groups (lines 17-20). The lesions were studied before and after treatment to create the study groups. The PRL group, and PRL1 and PRL2 subgroups are not explicit enough.

Response 1: Unfortunately, we have a limit of  200 words to compose the abstract and a lot of information had to be summarized. In lines 17 and 18 we briefly inform that the immunohistochemistry study was carried out with biopsies taken before the specific treatment. We were able to add to the definition of PRL1 that it was before the start of treatment (lines 19-20) and we also add the acronym TL (tegumentary leishmaniasis) in full (line 29-30).

Point 2: Despite the introduction being quite complete, the authors could still address the Th1, Th2 and regulatory immune response, because these concepts are extensively addressed in the discussion.

Response 2: Thank you for this suggestion. We agree. The literature establishes that the favorable immune response of patients with the localized cutaneous form (LCL) is characterized by Th1 cytokines, such as INF-γ, TNF-α (Tumor Necrosis Factor alpha) and interleukin 2. But a response with Th2 cytokines with the participation of IL-4, IL-5, IL-10, IL-13 and transforming growth factor-beta (TGFβ) may also be present. In order to include a brief information on this aim, we added the follow text: “In addition to the well-known influence of immune responses with predominance of Th1 T lymphocytes (favoring infection control) or Th2 (favoring potential parasite survival and consequently maintenance of active infection), other populations of lymphocytes such as T Helper 17 (Th17).... (lines 74 a 77).

Point 3: Throughout the text the same abbreviations are created several times and there are abbreviations without previous explanation/description (line 29). Please, correct.

Response 3: Thank you for pointing out this problem. The acronyms and their inclusions in the text were corrected.

Point 4: Line 24: The term "lower expression of macrophages" needs to be replaced by a more appropriate one, because "expression" has another biological meaning.

Response 4: We agree and we replaced the word expression by percentage.

Point 5:  Line 29: The abbreviation TL is used, but it was not explained.

Response 3: we add the acronym TL (tegumentary leishmaniasis) in full (line 29-30).

Point 6: Line 82: The reference is missing.

Response 6: Thank you, and we apologize for this mistake. We added the reference (31) to the text (line 87) and in the list of references (reference 31).

Point 7: Lines 82-83: Sentence needs improvement, particularly the expression “present difficulties”.

Response 7: Thank you for this question. In order to clarify we modify the text as follows: M1 favor parasite control while M2 cells favor parasite growth and survival, consequently, maintaining the lesion (lines 83-84).

Point 8: Line 152: Considering that FOXP3 has an intranuclear location, what precautions were taken during sample processing to ensure correct labeling?

Response 8: In flow cytometry studies, permeabilization is mandatory. In tissue immunohistochemistry studies, due to the tissue preparation process, fixatives, 3µm sections, etc., the cellular content is normally exposed and several articles published use the technique without permeabilization. Please find bellow some papers using FoxP3 staining by immunohistochemistry.

Xiaoxiao Li · Kai Ma · Shanai Song· Fangzhen Shen Tao Kuang· Yingqian Zhu· Zimin Liu. Tight correlation between FoxM1 and FoxP3+ Tregs in gastric cancer

and their clinical significance. Clinical and Experimental Medicine (2018) 18:413–420, https://doi.org/10.1007/s10238-018-0505-6.

Sonia Furgiuele, Géraldine Descamps, Jerome R. Lechien, Didier Dequanter, Fabrice Journe and Sven Saussez. Immunoscore Combining CD8, FoxP3, and CD68-Positive Cells Density and Distribution Predicts the Prognosis of Head and

Neck Cancer Patients. Cells 2022, 11, 2050. https://doi.org/10.3390/cells11132050

Point 9: Line 205: L. braziliensis

Response 9: Corrected.

Point 10: Table 1:  Table 1 appears to be not well formatted.

Response 10: We apologize, the table 1 has been formatted.

Point 11: Table 5: Table 5 has no caption.

Response 11: We apologize for this missing information. Caption was added as follows:

Lines 519– Table 5 - Panel of the in situ inflammatory profile according to each group evaluated.

Reviewer 4 Report

congratulate the Authors and thank them for their work.

I have just an observation: in line 82, I think you forgot to include the reference.

Author Response

ID: microorganisms-2311054 - “Is there any difference in the in situ immune response in active localized cutaneous leishmaniasis that respond well or poorly to meglumine antimoniate treatment or spontaneously heal?.”

Response to Reviewer 4 Comments

Point 1:   I have just an observation: in line 82, I think you forgot to include the reference.

Response 1: Thank you very much for reviewing the manuscript and for your positive feedback. We added the reference (31) to the text (line 87) and in the list of references (reference 31).
